# Soil Organic Carbon Sequestration and Active Carbon Component Changes Following Different Vegetation Restoration Ages on Severely Eroded Red Soils in Subtropical China

**Shengsheng Xiao** [1,2]**, Jie Zhang** [1,2]**, Jian Duan** [1,2]**, Hongguang Liu** [1,2]**, Cong Wang** [3]
**and Chongjun Tang** [1,2],*

1   Jiangxi Provincial Key Laboratory of Soil Erosion and Prevention, Nanchang 330029, China;
    xss19811213@163.com (S.X.); zhonglilinzi@126.com (J.Z.); djlynn20@126.com (J.D.);
    xinongfish@163.com (H.L.)
2   Jiangxi Institute of Soil and Water Conservation, Nanchang 330029, China
3   State Key Laboratory of Urban and Regional Ecology, Research Center for Eco-Environmental Sciences,
    Chinese Academy of Sciences, Beijing 100085, China; wang7088sdu@126.com
*   Correspondence: tangchongjun@126.com or tangchongjun18@mails.ucas.ac.cn

**Abstract:** Degraded soil has a high carbon sink potential. However, the carbon sequestration capacity and efficiency of comprehensive control measures in soil erosion areas are still not fully understood, and this information is essential for evaluating the effects of adopted restoration measures. The objective of this study was to determine the restoration of soil organic carbon and active carbon components under the impact of soil erosion measures and reforestation following different restoration ages. A small watershed with four typical restored plots following the same control measures (combination measures with horizontal bamboo burl-groove + replanting trees, shrubs and grasses) but different restoration ages (4 years, 14 years, 24 years and 34 years) and two reference plots (bare land (carbon-depleted) and nearby undisturbed forest (carbon-enriched)) in subtropical China was studied. The results showed that the soil organic carbon contents at a 1 m soil depth and the dissolved organic carbon and microbial biomass carbon concentrations in the upper 60 cm of soils of the four restored lands were higher than those in the bare land. Furthermore, the restored lands of 4 years, 14 years, 24 years and 34 years had soil organic carbon stocks in the 1 m soil depth of 22.83 t hm$^{-2}$, 21.87 t hm$^{-2}$, 32.77 t hm$^{-2}$ and 39.65 t hm$^{-2}$, respectively, which were higher than the bare land value of 19.86 t hm$^{-2}$ but lower than the undisturbed forestland value of 75.90 t hm$^{-2}$. The restored forestlands of 34 years of ecological restoration also had a high potential of being a soil organic carbon sink. Compared to the bare land, the restored lands of 4 years, 14 years, 24 years and 34 years had soil organic carbon sequestration capacities of 2.97 t hm$^{-2}$, 2.01 t hm$^{-2}$, 12.91 t hm$^{-2}$ and 19.79 t hm$^{-2}$, respectively, and had soil organic carbon sequestration rates of 0.74 t hm$^{-2}$ a$^{-1}$, 0.14 t hm$^{-2}$ a$^{-1}$, 0.54 t hm$^{-2}$ a$^{-1}$ and 0.58 t hm$^{-2}$ a$^{-1}$, respectively. Our results indicated that the combined measures of horizontal bamboo burl-groove and revegetation could greatly increase carbon sequestration and accumulation. Suitable microtopography modification and continuous organic carbon sources from vegetation are two main factors influencing soil organic carbon recovery. Combination measures, which can provide suitable topography and a continuous soil organic carbon supply, could be considered in treating degraded soils caused by water erosion in red soil areas.

**Keywords:** soil organic carbon; carbon sequestration; active organic carbon; eroded soils; ecological restoration

## 1. Introduction

Forest soils play a key role in the global carbon (C) cycle and the future mitigation of climate change because approximately 45% of terrestrial C is in forest soils [1]. The accumulated C pool in soils is more stable than the vegetation biomass that serves as a temporary C pool [2]. However, long-term severe human disturbance has had a serious effect on subtropical forest ecosystems, with complex topography and climate change resulting in fewer climax forests and a decrease in the functioning of an ecological security barrier [3]. As highlighted by the Bonn Challenge, forest vegetation restoration has become a priority research area in efforts to solve global environmental problems, e.g., the global effort to restore $1.50 \times 10^6$ km$^2$ of degraded land and deforested areas by 2020 [4].

The red soil hilly area in southern China covers an area of $1.18 \times 10^6$ km$^2$, of which the soil erosion area accounts for approximately 15%, becoming the second largest soil erosion area in China after the Loess Plateau [5]. In particular, the red soil developed by granite is seriously degraded, with a low C density and a high C sequestration potential [6,7]. Increasing the C sink of degraded soil has become a major problem to be solved in China. Many countries, including China, have established national programmes to increase vegetation areas [8]. The Chinese government initiated a series of state-funded forestry ecological projects, especially in relation to the key projects of soil and water conservation [9,10]. In the red soil erosion area of South China, the combined measures of horizontal bamboo burl-groove + replanting trees, shrubs and grasses is the most common control mode for treating soil erosion, especially for moderate and intensive erosion areas [11]. In the early 1980s, the soil erosion area of the southern part of Jiangxi Province (Ganzhou city) was approximately $1.10 \times 10^4$ km$^2$, of which 40% had an intensity that was moderate or greater. In the soil erosion area of moderate and above intensity, 60% of soil erosion areas were governed by the combination measures [12]. The combination measures had significant soil and water control effects, with surface runoff reductions of 70% and soil loss reductions of 80% [13]. Consequently, forest vegetation has been rapidly restored, forming a series of secondary vegetation communities at different restoration stages in this area.

By the C accumulation and storage in both plant biomass and soil, vegetation restoration is an effective method to alleviate the carbon dioxide ($CO_2$) concentration in the air [14,15]. Lal [16] reported that 60%–75% of the C loss in degraded soils could be re-fixed by ecological restoration, and the C sequestration potential of globally degraded soils was estimated to be $3 \times 10^8$–$8 \times 10^8$ t a$^{-1}$. In recent years, studies regarding the influence of afforestation/ecological restoration/land-use change on soil C stocks have been conducted extensively [17,18]. Numerous studies have demonstrated that C stocks in plant biomass and soil can rapidly increase with afforestation [19,20] and recover to an equivalent level to that of nearby undisturbed forests within a few decades [21], particularly in tropical regions, where consistent warm temperatures and ample rainfall favour rapid plant growth [22]. However, how forest restoration affects soil C recovery in eroded and degraded regions is still not well understood [23], especially for the severely eroded red soils in subtropical China [6], which leads the ignores of C sequestration benefits of soil and water conservation. Furthermore, in the process of vegetation restoration, the change in total soil organic carbon (TOC) accumulation cannot indicate the impact of ecological restoration on soil C dynamics. The identification of more sensitive active organic C components is helpful for clarifying the soil organic carbon (SOC) dynamics [24], though most previous studies on reforested lands have focused on C stock changes under land-use type transfers [6,25–27]. However, active organic C components and soil C turnover have largely been overlooked, and little attention has been given to the effects of restoration progress or levels of degraded soils on C sequestration [6,28]. This knowledge may have important implications for the quality of the soil C stock, storage time and long-term soil C sequestration in these restored forests, and we can evaluate the effects of decades of constructive efforts of restoration measures on degraded land [6,29].

The Tangbei River watershed was the first small watershed used for the comprehensive control of soil and water loss in the Yangtze River basin beginning in the early 1980s. In this study, the Tangbei River watershed and the Pingjiang River basin were chosen as the research sites. The objective of this

study was to evaluate the effects of vegetation restoration at different ages on soil C sequestration in an eroded degraded subtropical region of South China. From a chronosequence of adjacent forest sites, we addressed this research in bare land (BL) as a control and four restored forest sites with different restoration ages of 4 years (F4), 14 years (F14), 24 years (F24) and 34 years (F34). A nearby-undisturbed forest (UF, representing the climax vegetation) was also selected to test the recovered level of the soil C stock after long-tern vegetation restoration. With the continued input of plant litter, we hypothesized long-term vegetation restoration would increase the soil C stocks. We also hypothesized that, due to the restored forest age was still short relative to the age of the primary forests, the soil C stocks in the F34 sites would be lower than those in the UF sites. Finally, we hypothesized that the soil active organic C content gradually increased with the increase in recover age (in years), which increased the risk of soil C decomposition and decreased the stability of soil C stocks. Therefore, compared to the early recovery stages, the increase rate of soil C stocks in the late recovery period will be reduced.

## 2. Material and Methods

### 2.1. Study Site Description

The study was conducted in the Pingjiang Basin (Figure 1), Jiangxi Province, which is in the central region of the red soil hilly region in South China. The study area elevation ranges from 127 m to 1207 m and is located in a subtropical humid monsoon climate zone, with an average annual temperature of 18.9 °C and an average annual precipitation of 1538.7 mm (48.7% of which occurs from April to June). The zonal soil is made up of Quaternary red soil with a sandy and clay loam texture and poor corrosion resistance.

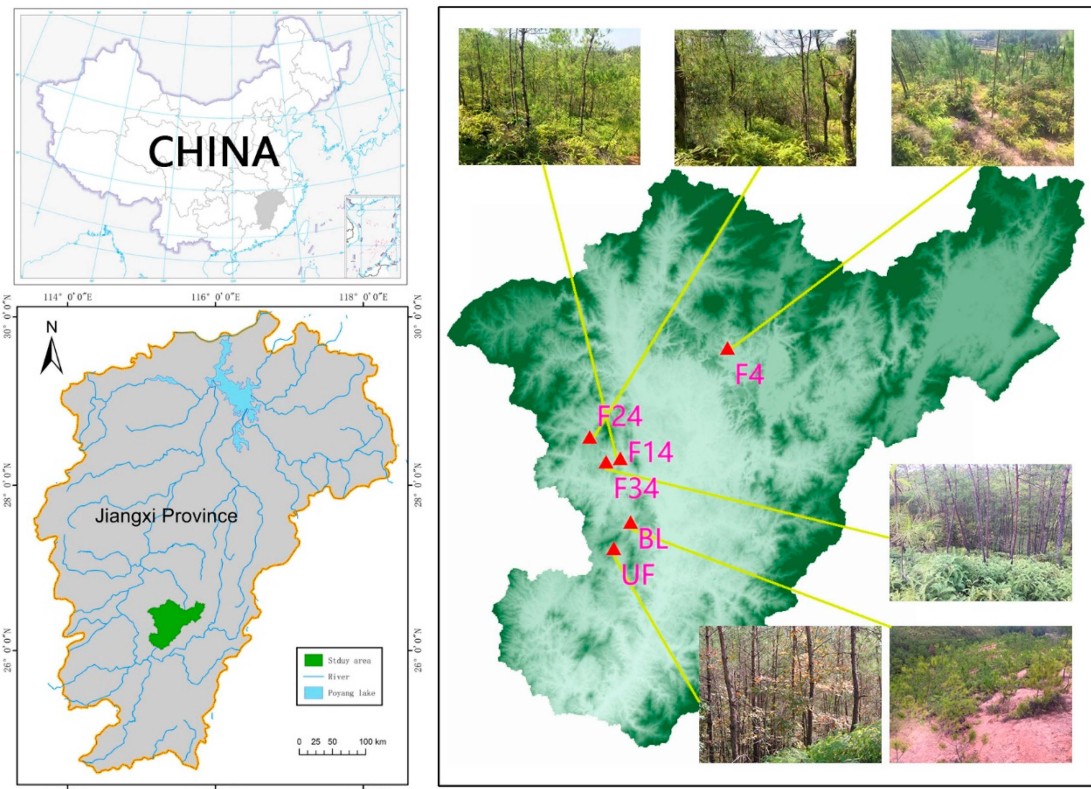

**Figure 1.** Location of study area and the distributions of six forest sites. BL: Bare land; F4, F14, F24, and F34: the forest sites of 4 years, 14 years, 24 years, and 34 years since the completion of combined measures; UF: nearly undisturbed old-growth forest sites.

In the Pingjiang Basin, the original evergreen broad-leaved forests were seriously damaged. The existing vegetation is mainly *Pinus massoniana* secondary forest and artificial forest (citrus orchard,

Chinese fir, etc.). Long-term human activities have resulted in severe soil erosion. In the early 1980s, the average vegetation coverage of the Tangbei River watershed was 28.8%, with the proportion of intensive loss hilly areas at only 10%, now known as the "red desert of southern China" [30]. After long-term ecological construction efforts before 2015, the vegetation coverage was approximately 85% [31].

In the process of ecological construction, the main measure mode was horizontal bamboo burl-groove + replanting trees, shrubs and grasses. Horizontal bamboo burl-groove is a kind of groove built on the hillside at regular intervals along the contour line for water storage and sediment retention. A soil barrier is set at a certain distance (approximately 1.5 m) in the groove to stop the flow of water, similar to a bamboo burl. The common design specifications are trapezoidal sections that are 0.5 m deep, 0.6 m wide at the upper edge and 0.4 m wide at the bottom. The width and depth of the bamboo burl are 0.1 m and 0.3 m, respectively. The excavated soil is used as a ridge outside the ditch with a width of 0.25 m. The distance between two grooves is 3–5 m, according to the slope and erosion degree. Before ecological construction, the vegetation structure was very simple, with some "old-pine" (*Pinus massoniana* Lamb.) and some *Arundinella setosa*, *Eriachne pallescens* R. Br. The replanting plants are *Liquidambar formosana* Hance, *Schima superba* Gardn. et Champ., *Lespedeza bicolor* Turcz. and *Paspalum wettsteinii* Hack.

In this study, six forest sites with three replicates were chosen (Figure 1). Three plots (20 m × 20 m) were randomly set up at each stand site. First, the BL sites were used as a control, representing the original state before restoration. Due to the complete erosion of topsoil, coarse sands and mineral aggregates rich in Fe and Mn covered the bare surface. Second, we followed the control process of red soil eroded areas and selected four main stages of vegetation restoration, F4, F14, F24, and F34 (referring to 4 years, 14 years, 24 years, and 34 years since the completion of soil erosion mitigation efforts, respectively); the sites had the same parent rock and similar topographical conditions. Before soil erosion control, the entire Ah layer of the four forest sites had been lost due to soil erosion. From the end of soil erosion measures to now, all sites have been closed with no human disturbance. Finally, a nearby undisturbed forest (UF) represents the climax vegetation. The main characteristics of the vegetation and the surface soils (0–20 cm) among the six research sites are shown in Tables 1 and 2.

## 2.2. Soil Sample Collection and Determination

Soil samples were collected in December 2019 at five soil depths (0–10 cm, 10–20 cm, 20–40 cm, 40–70 cm and 70–100 cm). In each forest sites, five core samples were randomly collected and combined into one sample at different layers. Cut ring samples (100 cm$^3$ core volume) at each depth were also obtained for the determination of soil bulk density. After each sampling, the soil samples were placed in a cooler, transported back to the laboratory, and stored in a refrigerator at −4 °C. Fresh soil samples were sieved through a 2-mm mesh, with visible roots, organic debris, and rocks removed, and then the samples were divided into two subsamples. One subsample was air-dried to determine the content of TOC, $w$(TOC). The other subsample was stored at −4 °C for the determination of the content of soil dissolved organic carbon, $w$(DOC), and the content of microbial biomass carbon, $w$(MBC), which occurred no more than two weeks after sampling.

The samples were analysed for TOC using the $K_2Cr_2O_7$ oxidation method [32]. Soil DOC was extracted from 10 g of fresh soil with 50 mL of deionized water. After vibrating (at least 30 min), centrifugating (4000 r min$^{-1}$ for 20 min), and filtrating through a membrane filter with 0.45-μm pores, the $w$(DOC) was determined using the TOC analyser (Vario TOC Cube, Elementar, Germany) [33]. Soil MBC was extracted using the chloroform fumigation method [33]. The 0.5 mol L$^{-1}$ $K_2SO_4$ solution was used to extract C from fumigated and nonfumigated samples at a 1:10 (w:v) ratio and then quantified by a TOC analyser within a week after sampling.

**Table 1.** Summary of the vegetation characteristics of the six research forest sites. Data are mean ± SD.

| Research Site | Longitude and Latitude | Species Richness | Simpson Index | Vegetation Cover | | | Most Abundant Species | | | Main Characteristics of *Pinus massoniana* Lamb. | |
|---|---|---|---|---|---|---|---|---|---|---|---|
| | | | | Trees | Shrubs | Herbaceous | Trees | Shrubs | Herbaceous | Average Tree Height (m) | Average DBH (cm) |
| BL | E 115.27333 N 26.25305 | 3.67 ± 1.53 | 0.51 ± 0.20 | 28.36% ± 16.07% | 0 | 22.00% ± 2.65% | *Pinus massoniana* Lamb. | / | *Dicranopteris dichotoma* (Thunb.) Berhn. | 1.15 ± 0.14 | 2.15 ± 0.22 |
| F4 | E 115.41120 N 26.42831 | 10.67 ± 0.58 | 0.82 ± 0.02 | 13.67% ± 3.79% | 3.00% ± 1.00% | 67.22% ± 19.67% | *Pinus massoniana* Lamb. | *Rhododendron simsii* Planch. | *Dicranopteris dichotoma* (Thunb.) Berhn. | 1.78 ± 0.16 | 2.68 ± 1.13 |
| F14 | E 115.26878 N 26.32236 | 6.12 ± 2.05 | 0.68 ± 0.06 | 32.30% ± 11.15% | 0.78% ± 0.88% | 81.33% ± 11.61% | *Pinus massoniana* Lamb. | *Camellia oleifera* Abel, *Lespedeza bicolor* Turcz | *Dicranopteris dichotoma* (Thunb.) Berhn. | 3.01 ± 0.68 | 4.38 ± 0.80 |
| F24 | E 115.23506 N 26.34845 | 5.67 ± 2.52 | 0.63 ± 0.06 | 64.67% ± 12.22% | 0.53% ± 0.92% | 85.83% ± 9.97% | *Pinus massoniana* Lamb. | *Camellia oleifera* Abel | *Dicranopteris dichotoma* (Thunb.) Berhn. | 4.09 ± 0.55 | 4.43 ± 0.86 |
| F34 | E 115.26872 N 26.32231 | 8.00 ± 2.65 | 0.71 ± 0.05 | 43.33% ± 6.11% | 2.04% ± 1.58% | 90.78% ± 3.74% | *Pinus massoniana* Lamb. | *Camellia oleifera* Abel, *Lespedeza bicolour* Turcz | *Dicranopteris dichotoma* (Thunb.) Berhn. | 5.99 ± 1.04 | 7.14 ± 0.56 |
| UF | E 115.24967 N 26.22693 | 9.03 ± 4.00 | 0.84 ± 0.08 | 92.67% ± 4.62% | 10.08% ± 3.63% | 7.94% ± 4.90% | *Castanopsis carlesii* (Hemsl.) Hay., *Cunninghamia lanceolata* (Lamb.) Hook., *Cyclobalanopsis glauca* (Thunberg) Oerste | *Loropetalum chinensis* (R. Br.) Oliv. | *Dicranopteris dichotoma* (Thunb.) Berhn. | / | / |

**Table 2.** Soil physical and chemical characteristics of the surface soils (0–20 cm) among six research sites. Note: Data are mean ± SD, $n = 3$; Different letters indicate significant differences among different research sites ($p < 0.05$). BL: Bare land; F4, F14, F24, and F34: the forest sites of 4 years, 14 years, 24 years, and 34 years since the completion of combined measures, respectively; UF: nearly undisturbed old-growth forest.

| Research Site | Soil Texture (Sand-Silt-Clay, %) | Bulk Density (BD, g cm⁻³) | pH (1:2.5) | Total Nitrogen (TN, g kg⁻¹) | Available Nitrogen (AN, mg kg⁻¹) | Total Phosphorus (TP, g kg⁻¹) | Available phosphorus (AP, mg kg⁻¹) | Maximum Water Holding Capacity (MWH, %) | Field water Holding Capacity (FWH, %) |
|---|---|---|---|---|---|---|---|---|---|
| BL | - | 1.47 ± 0.03a | 4.81 ± 0.11b | 0.20 ± 0.03b | 5.79 ± 1.91b | 0.08 ± 0.01d | 1.11 ± 0.91b | 31.44 ± 1.60d | 25.43 ± 1.62c |
| F4 | 57.92-29.70-12.38 | 1.30 ± 0.03b | 4.81 ± 0.08b | 0.21 ± 0.03b | 8.29 ± 0.76b | 0.14 ± 0.02c | 1.52 ± 0.28b | 40.97 ± 2.54bc | 33.73 ± 0.36b |
| F14 | 57.72-26.08-16.20 | 1.06 ± 0.09c | 5.10 ± 0.07a | 0.21 ± 0.07b | 7.94 ± 3.39b | 0.16 ± 0.02c | 1.88 ± 0.09b | 50.82 ± 3.70a | 38.19 ± 0.77a |
| F24 | 65.44-22.40-12.17 | 1.23 ± 0.02b | 4.80 ± 0.03b | 0.26 ± 0.04b | 6.95 ± 2.55b | 0.15 ± 0.01c | 1.27 ± 0.30b | 40.90 ± 0.82bc | 32.97 ± 0.63b |
| F34 | 71.12-18.61-10.27 | 1.11 ± 0.04c | 4.71 ± 0.05bc | 0.26 ± 0.04b | 6.22 ± 1.60b | 0.26 ± 0.02b | 1.85 ± 0.21b | 45.22 ± 2.29b | 32.73 ± 3.40b |
| UF | 57.56-20.66-21.78 | 1.23 ± 0.08b | 4.61 ± 0.09c | 1.68 ± 0.16a | 76.96 ± 12.61a | 0.39 ± 0.07a | 4.98 ± 3.69a | 37.28 ± 2.95c | 28.39 ± 2.44c |

*2.3. Data Processing and Statistical Analysis*

SOC stocks were calculated for the 0–100 cm soil layer using Equation (1) [34]:

$$SOC_{stocks} = \sum_{i=1}^{n} D \times BD_i \times (1 - f) \times SOC_i \qquad (1)$$

where $n$ is the number of soil layers; $D$, $BD_i$, $f$ and $SOC_i$ represent the thickness of the soil layer (cm), the bulk density (g cm$^{-3}$), the gravel content in each layer of soil (>2 mm, %), and the $w$(TOC) of soil layer $i$, respectively.

The relative soil C sequestration capacity (t hm$^{-2}$) was adapted from the changes in the SOC stocks between the restored forest sites and the BL site [24,35]. Therefore, the soil C sequestration rate (t hm$^{-2}$ a$^{-1}$) could be reasonably calculated using the soil C sequestration capacity and restoration period.

All ANOVA and correlation analyses were conducted with a significance criterion of $p < 0.05$ (unless otherwise declared) using IBM SPSS 19 statistical software (SPSS, Chicago, IL, USA), and graphs were prepared with Origin 8.5 (Origin Lab Corporation, Northampton, MA, USA). All the results are presented as the mean ± SD. Before performing ANOVA, the normality of distribution was checked by a Kolmogorov-Smirnov test, and homogeneity of variance was determined by Levene's test. All data passed these tests, and no transformation was needed. The physical-chemical properties of surface soils for each plot were the mean values of the replicate samples. The concentrations of SOC, DOC, MBC and SOC stocks with respect to soil depth for each plot were the mean values of the three different topographic positions. Differences in the $w$(TOC), $w$(DOC), $w$(MBC) and soil variables among the different forest sites were analysed using one-way ANOVA with Tukey's HSD test.

## 3. Results

*3.1. Soil C Stocks*

The average TOC content ($w$(TOC)) at the 1 m soil depth was 2.26 g kg$^{-1}$ at the BL site. After the same control measurements for soil erosion with different restoration stages, $w$(TOC) mostly increased (except at the F4 sites, Figure 2). Compared with the BL, the average $w$(TOC) increased by 33.69%, 70.07% and 145.46% at the F14, F24 and F34 sites, respectively, reaching 3.02 g kg$^{-1}$, 3.84 g kg$^{-1}$ and 5.54 g kg$^{-1}$, respectively. At site F4, the $w$(TOC) decreased by 7.97% compared to BL, which may be due to human disturbance, causing bamboo-burl-groove construction in the early restoration stage. The nearby UF forest sites had the highest $w$(TOC) of 10.22 g kg$^{-1}$. This result means that after 34 years of restoration, the $w$(TOC) still had a large increase space for eroded red soil in subtropical China.

The vertical distribution of $w$(TOC) was obviously different in the different forest sites. In the BL and F4 plots, different soil layers had the same $w$(TOC) with variation coefficients of 25.55% and 31.35%, respectively ($p > 0.05$). In the F14, F24, F34 and UF sites, the $w$(TOC) in the 0–10 cm layer was significantly higher than that in the other soil layers ($p < 0.05$). In F24 and F34, the second soil layers (10–20 cm) had significantly higher $w$(TOC) than the other deep soils ($p < 0.05$). In the UF plots, there were significant differences among the five soil layers ($p < 0.05$). Consequently, the effect of vegetation restoration on TOC in deep soil was relatively small.

Because of long-term soil erosion and mineralization, the BL sites had the lowest SOC stocks of 19.86 t hm$^{-2}$ (Table 3). The SOC stocks of F4, F14, F24 and F34 were 22.83 t hm$^{-2}$, 21.87 t hm$^{-2}$, 32.77 t hm$^{-2}$ and 39.65 t hm$^{-2}$, respectively. Compared to BL sites, ecological restoration can significantly increase SOC storage, and SOC stocks increased with the restoration time (except for the F14 sites, maybe because of the soil sample collection). The proportion of SOC stocks in topsoil (0–20 cm) in the total 1 m soil depth in the BL sites was 27.38%. With the extension of recovery time, the proportion increased (except in the F4 site).

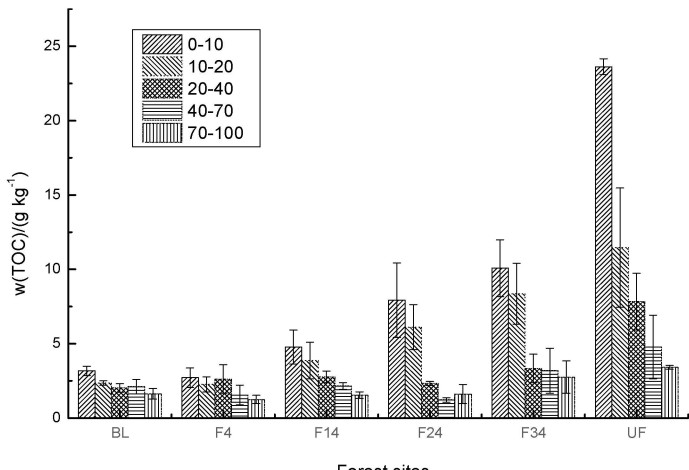

**Figure 2.** The depth distribution of SOC concentration in different forest sites. Values are mean ± SD, $n = 3$; BL: Bare land; F4, F14, F24, and F34: the forest sites of 4 years, 14 years, 24 years, and 34 years since the completion of combined measures; UF: nearly undisturbed old-growth forest sites.

### 3.2. DOC and MBC

The average DOC contents ($w$(DOC)) in the topsoil layers increased in the early recovery period (4 years and 14 years) but did not reach a significant level ($p > 0.05$, Table 4). With the extension of recovery time, compared to BL and F4, there were significant increases of $w$(DOC) in F24, F34 and UF ($p < 0.05$). The $w$(DOC) presented no significant difference among the F24, F34 and UF sites ($p > 0.05$). The change laws were reflected in the three soil layers. The $w$(DOC) in different soil layers had an obvious change rule. In the BL sites, the average $w$(DOC) in the 0–10 cm soil layer was significantly higher than that in the 10–20 cm and 20–40 cm soil layers ($p < 0.05$), and there was no significant difference in the average $w$(DOC) in the 10–20 cm and 20–40 cm soil layers ($p > 0.05$). After vegetation restoration, at every recovery stage, the significant differences between soil layers were eliminated ($p > 0.05$).

After vegetation restoration, the MBC contents ($w$(MBC)) in the three soil layers increased (Table 4). Similar to the DOC, $w$(MBC) did not increase significantly in the early recovery stages (in the F4 and F14 sites). With the extension of recovery time, the $w$(MBC) increased faster. Compared to the BL sites, the $w$(MBC) presented significant increases in the F24, F34 and UF sites, both for the 0–10 cm and for the 20–40 cm soil layers ($p < 0.05$). Different from DOC, there was no significant difference among different soil layers in most sites ($p < 0.05$), including the severely eroded sites, the four restoration age sites and the undisturbed forest sites (except between the 0–10 cm and 20–40 cm layers in the F24 sites).

In general, the ratios of $w$(DOC) and $w$(MBC) to $w$(TOC) in upper layer soil were both at a low level (Table 5). The average ratios of the upper three layers of $w$(DOC) to $w$(TOC) were 0.65%, 1.08%, 0.75%, 1.07%, 0.70% and 0.42% in the BL, F4, F14, F24, F34 and UF sites, respectively. The average proportions of the upper three layers of $w$(MBC) to $w$(TOC) were 1.44%, 1.75%, 1.63%, 2.84%, 1.93% and 1.90% in the BL, F4, F14, F24, F34 and UF sites, respectively. There was no significant difference between them ($p > 0.05$). Vegetation restoration had no significant effect on the ratios of $w$(DOC) and $w$(MBC) to $w$(TOC) in this study.

**Table 3.** The SOC storage and proportions of different soil layers to the 1 m depth soil profile.

| Soil Layer/cm | BL | | F4 | | F14 | | F24 | | F34 | | UF | |
|---|---|---|---|---|---|---|---|---|---|---|---|---|
| | Soil C Stocks (t hm⁻²) | Proportion (%) | Soil C Stocks (t hm⁻²) | Proportion (%) | Soil C Stocks (t hm⁻²) | Proportion (%) | Soil C Stocks (t hm⁻²) | Proportion (%) | Soil C Stocks (t hm⁻²) | Proportion (%) | Soil C Stocks (t hm⁻²) | Proportion (%) |
| 0–10 | $3.15 \pm 0.40$ | 15.85% | $3.27 \pm 0.76$ | 14.34% | $3.33 \pm 0.84$ | 15.21% | $8.79 \pm 2.64$ | 26.83% | $7.81 \pm 1.12$ | 19.69% | $21.31 \pm 1.26$ | 28.08% |
| 10–20 | $2.29 \pm 0.16$ | 11.53% | $2.67 \pm 0.58$ | 11.69% | $3.11 \pm 1.07$ | 14.22% | $7.65 \pm 1.87$ | 23.33% | $8.22 \pm 2.20$ | 20.72% | $10.23 \pm 3.12$ | 13.48% |
| 20–40 | $3.54 \pm 0.45$ | 17.83% | $6.21 \pm 2.22$ | 27.20% | $4.88 \pm 0.74$ | 22.30% | $5.90 \pm 0.29$ | 18.00% | $5.88 \pm 1.28$ | 14.82% | $16.45 \pm 4.64$ | 21.68% |
| 40–70 | $6.16 \pm 1.44$ | 31.02% | $6.05 \pm 2.62$ | 26.50% | $6.55 \pm 0.75$ | 29.95% | $4.54 \pm 0.62$ | 13.86% | $8.97 \pm 4.23$ | 22.63% | $16.23 \pm 9.17$ | 21.39% |
| 70–100 | $4.72 \pm 0.91$ | 23.77% | $4.63 \pm 1.04$ | 20.29% | $4.01 \pm 0.31$ | 18.33% | $5.89 \pm 2.68$ | 17.99% | $8.78 \pm 3.50$ | 22.14% | $11.67 \pm 0.30$ | 15.37% |
| In total | 19.86 | 100% | 22.83 | 100% | 21.87 | 100% | 32.77 | 100% | 39.65 | 100% | 75.90 | 100% |

**Table 4.** Vertical distribution of $w$(DOC) and $w$(MBC) in the different forest sites.

| Soil Layer/cm | $w$(DOC) (mg kg⁻¹) | | | | | | $w$(MBC) (mg kg⁻¹) | | | | | |
|---|---|---|---|---|---|---|---|---|---|---|---|---|
| | BL | F4 | F14 | F24 | F34 | UF | BL | F4 | F14 | F24 | F34 | UF |
| 0–10 | $20.79 \pm 1.11Ab$ | $32.14 \pm 8.64Ab$ | $32.49 \pm 10.05Ab$ | $58.79 \pm 11.25Aa$ | $52.21 \pm 9.44ABa$ | $58.04 \pm 13.24b$ | $32.47 \pm 19.01Ac$ | $47.33 \pm 36.53Ac$ | $73.85 \pm 11.01Abc$ | $199.43 \pm 61.47Ab$ | $142.57 \pm 93.22Abc$ | $349.70 \pm 140.98Aa$ |
| 10–20 | $11.94 \pm 2.18Bc$ | $27.43 \pm 11.26Ab$ | $31.92 \pm 6.96Aab$ | $52.53 \pm 23.77Aab$ | $56.59 \pm 15.13Aa$ | $52.72 \pm 15.44a$ | $49.23 \pm 21.45Ab$ | $46.71 \pm 19.44Ab$ | $51.89 \pm 24.97Ab$ | $112.93 \pm 59.07ABb$ | $80.50 \pm 6.57Ab$ | $236.96 \pm 83.78Aa$ |
| 20–40 | $15.76 \pm 2.76Bc$ | $22.21 \pm 6.37Ab$ | $20.86 \pm 7.54Ab$ | $37.03 \pm 5.24Aa$ | $30.07 \pm 8.30Bab$ | $43.00 \pm 13.09a$ | $24.58 \pm 6.35Ac$ | $38.08 \pm 12.22Ac$ | $54.80 \pm 18.93Ac$ | $96.48 \pm 26.34Bb$ | $113.96 \pm 35.55Ab$ | $169.31 \pm 19.99Aa$ |

Notes: Data are the mean $\pm$ SD, $n = 3$. Different capital letters in the same column indicate significant differences under different soil layers at one research site ($p < 0.05$). Different lowercase letters in the same row indicate significant differences among different research sites in the same soil layer ($p < 0.05$).

**Table 5.** The ratios of $w$(DOC) and $w$(MBC) to $w$(TOC) (%).

| Soil Layer/cm | DOC | | | | | | MBC | | | | | |
|---|---|---|---|---|---|---|---|---|---|---|---|---|
| | BL | F4 | F14 | F24 | F34 | UF | BL | F4 | F14 | F24 | F34 | UF |
| 0–10 | 0.66% | 1.19% | 0.68% | 0.74% | 0.52% | 0.25% | 1.02% | 1.75% | 1.55% | 2.52% | 1.41% | 1.48% |
| 10–20 | 0.51% | 1.21% | 0.83% | 0.86% | 0.68% | 0.46% | 2.08% | 2.07% | 1.34% | 1.85% | 0.96% | 2.07% |
| 20–40 | 0.78% | 0.85% | 0.75% | 1.59% | 0.90% | 0.55% | 1.21% | 1.45% | 1.98% | 4.15% | 3.42% | 2.16% |

### 3.3. Relationship between w(TOC), w(DOC), w(MBC) and Soil Properties

There was a significant linear positive correlation between $w$(TOC) and $w$(TN), $w$(AN) ($p < 0.001$) and $w$(TP) ($p < 0.05$) in this study (Table 6). $w$(DOC) had significantly positive correlations with $w$(TN) ($p < 0.05$) and $w$(TP) ($p < 0.001$). $w$(MBC) was linear positively correlated to the content of TN, AN, TP at 0.001 level and AP at the level of 0.05, and linear negatively correlated to the soil pH (Table 6). Therefore, besides the SOC pool, ecological restoration also had a significant impact on soil N and soil P pool. Pearson's correlation analysis showed that there was a certain coupling between SOC and soil N and P. Considering the close relationship between soil microorganisms and active soil C, it is necessary to strengthen the investigation of soil microbial communities in different restoration stages.

**Table 6.** Pearson's correlation between TOC, DOC, MBC and soil properties.

| | pH | BD | TN | AN | TP | AP | FWH | MWH | TOC | DOC | MBC |
|---|---|---|---|---|---|---|---|---|---|---|---|
| TOC | −0.252 | −0.311 | 0.735 ** | 0.784 ** | 0.543 * | 0.443 | −0.186 | −0.153 | 1 | 0.411 | 0.728 ** |
| DOC | −0.444 | −0.443 | 0.484 * | 0.462 | 0.641 ** | 0.368 | 0.095 | 0.142 | 0.411 | 1 | 0.760 ** |
| MBC | −0.517 * | −0.223 | 0.814 ** | 0.820 ** | 0.765 ** | 0.544 * | −0.217 | −0.107 | 0.728 ** | 0.760 ** | 1 |

$* p < 0.05$, $** p < 0.001$. BD: Bulk density; TN: Total nitrogen; AN: Available nitrogen; TP: Total phosphorus; AP: Available phosphorus; MWH: Maximum water holding capacity; FWH: Field water holding capacity.

### 3.4. SOC Sequestration Rate and Potential

Compared to the BL, the soil C stocks in 1 m depth soil for restored forest sites significantly increased. Vegetation growth in response to ecological restoration has led to improved ecosystem services, including SOC sequestration. Compared to the BL sites, the SOC sequestration capacities of the F4, F14, F24 and F34 sites were 2.97 t hm$^{-2}$, 2.01 t hm$^{-2}$, 12.91 t hm$^{-2}$ and 19.79 t hm$^{-2}$, respectively. Therefore, compared to the BL sites, after different restoration ages, the SOC sequestration rates of the F4, F14, F24 and F34 sites were 0.74 t hm$^{-2}$ a$^{-1}$, 0.14 t hm$^{-2}$ a$^{-1}$, 0.54 t hm$^{-2}$ a$^{-1}$ and 0.58 t hm$^{-2}$ a$^{-1}$, respectively. Therefore, this study showed that in the typical red soil area in subtropical China, even after 34 years of ecological restoration, the forest soil still maintained a high SOC sequestration rate. However, the SOC increase rates differed with restoration time. In the recovery later stage, the SOC sequestration rates were higher than those in the early stages.

In our study area, the average SOC stocks in the UF sites was 75.90 t hm$^{-2}$. Consequently, compared to the UF sites, the SOC sink potentials of BL, F4, F14, F24 and F34 were 56.04 t hm$^{-2}$, 53.07 t hm$^{-2}$, 54.03 t hm$^{-2}$, 32.77 t hm$^{-2}$ and 36.25 t hm$^{-2}$, respectively. Additionally, after 34 years of ecological restoration, the F34 sites had a high potential to sequester SOC. Assuming that the F34 sites had an SOC sequestration rate of 0.58 t hm$^{-2}$ a$^{-1}$ later, 62 years are needed to reach the equivalent SOC storage level of the nearby undisturbed secondary forest.

## 4. Discussion

### 4.1. Soil C Stock Recovery after Ecological Restoration

Soil C dynamics are controlled by the complex interplay of climatic, edaphic, and biotic factors. In agreement with our hypothesis, vegetation restoration in the subtropical eroded red soil region significantly increased the soil C stocks. Many previous studies have obtained similar conclusions [6,25,26].

In our study, the recovery of soil C stocks after vegetation restoration may be mainly due to the input of easy decomposed plant litter into the soil following the establishment and growth of trees, shrubs and herbs [36] and the enhanced physical protection of soil aggregates [37]. The annual input of plant and root residues is considered to be the major source of SOC for degraded soils [6,38,39]. It should be noted that decreased soil erosion under the combined measures reduced the losses of soil C and may have partly facilitated the accumulation of SOC. The revegetation (replanting trees, shrubs and grasses) worked mainly as a carbon source, and the horizontal bamboo burl-groove changed the topography to prevent soil erosion. In addition, after vegetation restoration, the slow decomposition rate of old soil C was an important reason for the increase in SOC in the red soil erosion area of southern China [40]. Lu et al. [41] reported that in the process of vegetation restoration in red soil eroded land, unprotected SOC could gradually transform into protected SOC.

The effects of vegetation restoration on soil C sequestration remain controversial [40,42]. These inconsistent conclusions may be mainly due to different degradation degrees and soil nutrient levels (especially the limitation of soil N and P contents), and afforestation time [17,38,42].

Generally, without human disturbance, SOC will accumulate over time, reach a saturated level and tend to be stable [43]. Therefore, recovery age may play a key role in the increase in soil C stocks. Brown and Lugo [44] found that there was a general pattern of increasing soil C stocks with increasing age of secondary forests, and the 50-year-old secondary forests had approximately the same soil C stocks as the primary forests in the subtropics. Veloso et al. [18] stated that reforestation could restore the initial soil C stock relative to a subtropical natural forest after 32 years. In the eroded red soil region, Xie et al. [45] reported that it takes approximately 30–90 years for the SOC of *Pinus massoniana* forest to recover to the level of secondary forest soil, and it takes 56–170 years for the SOC in BL to recover to the level of secondary forest soil. Therefore, our study demonstrated the advantages of using a combined approach in treating degraded lands.

### 4.2. Soil Active C Changes after Vegetation Restoration

DOC and MBC are labile fractions of SOC and are highly sensitive to environmental changes. In this study, $w$(DOC) and $w$(MBC) showed linear increases during the restoration progress, which were consistent with the results of Trigalet [46]. This result tends to support the conclusion that vegetation type controls the availability of SOC [47]. Vegetation restoration re-established the material circulation pathway of litter and fine roots, which provided a material source for active OC. Fontaine et al. [48] also stated that the supply of fresh C may promote the decomposition of SOC and facilitate increases in DOC. The enhanced MBC content could be due to the plant residue inputs with afforestation [49]. However, some studies reported that the active OC would increase in the early stage of recovery and maintain a stable level in the late recovery stage [50] or decrease [27,41]. Therefore, the $w$(DOC) and $w$(MBC) have not reached saturation levels after 34 years following the combination measures in our study, and monitoring needs to continue in the subsequent recovery time.

However, whether the accumulated C after ecological restoration can be stored in the soil in the long term largely depends on SOC stabilization [51,52], especially under disturbance or future climate change [53]. Thus, changes in both the soil C stock and its stabilization determine the real capacity of C sequestration in forest soil. Furthermore, it is noteworthy that the MBC, which is related to soil microbial activity and soil microorganisms, played an important role in the SOC cycles [54]. The recovery of soil active OC during vegetation restoration increased the risk of SOC decomposition and release. Similar to the research conducted by Zhang et al. [24], although vegetation restoration had a significant C sequestration benefit, it reduced the stability of SOC. Xiong et al. [55] also demonstrated that forest restoration could increase soil C stocks equivalent to undisturbed old-growth forests within a few decades, but the rate of soil C turnover in these restored forests was still higher. In this research, the indicators of DOC and MBC for SOC turnover do not fully reflect SOC stability. In the future, more attention should be paid to the stability of different components of SOC, such as liable C pools and

recalcitrant C pools, which are important for the evaluation of the quality of soil C stock, storage time and long-term soil C sequestration in these restored forests [56,57].

*4.3. Soil C Sequestration Rate, Potential and Implications for Forest Management In Eroded Red Soil Regions*

In our study, compared to the BL sites, the SOC sequestration rates of F4, F14, F24 and F34 sites were 0.74 t hm$^{-2}$ a$^{-1}$, 0.14 t hm$^{-2}$ a$^{-1}$, 0.54 t hm$^{-2}$ a$^{-1}$ and 0.58 t hm$^{-2}$ a$^{-1}$, respectively. This result is in agreement with one previous study [58], which showed that the average SOC sequestration rate of 30-year-old *Pinus massoniana* forests established on bare land in the red soil region of southern China was 0.39 t hm$^{-2}$ a$^{-1}$. Post [36] also stated that restorative forests had an average SOC sequestration rate of 0.3–0.6 t hm$^{-2}$ a$^{-1}$ when transferred from abandoned farmland. This result confirms our hypothesis that the eroded soil had very large SOC sequestration potential and that the use of combined measures on degraded soil resulted in a high SOC sequestration capacity. Moreover, Yu et al. [59] demonstrated that there was high CO$_2$ uptake by subtropical forest ecosystems in the East Asian monsoon region with a net ecosystem productivity (NEP) of 362 ± 39 g C cm$^{-2}$ a$^{-1}$. Our results confirm this conclusion to some extent.

The SOC sequestration rate usually depends on the comprehensive effect of climate, soil, tree species, litter properties and other factors, which means that the fixed C in soil is not infinitely increased, and there is a maximum accumulation and C saturation level [60]. In our study area, after 34 years following the use of combined measures, there was still a large gap in the SOC stock between the F34 sites (39.65 t hm$^{-2}$) and UF sites (75.90 t hm$^{-2}$). Compared to UF, F34 had a SOC sequestration potential of 36.25 t hm$^{-2}$. This result is in agreement with previous studies. By meta-analysis, Don et al. [19] suggested that soil C stock level after vegetation restoration were still lower than those in the undisturbed forests. Zhou et al. [61] suggested that even aged forest soil (0–20 cm) with tree ages over 400 years in South China had a high SOC sequestration capacity of 0.61 t hm$^{-2}$ a$^{-1}$. Yang et al. [62] also showed that the SOC sequestration potential of degraded land in the mid-subtropical mountainous areas of China was lower than that in other regions at the same latitude, which was mainly due to precipitation and geomorphic conditions, especially C loss with slope runoff, which accounted for a large proportion. Some studies reported some restoration projects that achieved very limited success in SOC recovery, mostly due to nutrient limitations, poor tree species selection, and a lack of understanding of the plant-soil-microbe interaction [63–65].

In the future, when quantitatively evaluating the effects of ecological restoration on SOC storage in eroded regions, comprehensive research on key processes of the C cycle needs to be strengthened, including C loss by erosion, C release by soil respiration, and plant biomass C (including aboveground and underground biomass C). The increase in SOC due to the recovery of eroded soil will increase the risk of loss of SOC by soil erosion, especially in the red soil region of southern China. Moreover, the input of plant residues and the improvement of the soil environment after vegetation restoration, especially the recovery of active SOC, may promote the decomposition of SOM and soil respiration and may hence offset a small part of recovered soil C [66]. The comprehensive analysis of these factors will improve our understanding of the mechanisms of SOC change and sequestration.

The SOC recovery of degraded soil requires a long time. Our study may provide some significant implications for forest management in eroded red soil regions. First, more attention should be paid to the protection of secondary forests and restored vegetation in the red soil area of South China to avoid serious soil erosion caused by excessive human disturbance. For example, Jackson et al. [67] reported that the increase in aboveground biomass caused by trees and shrubs might be offset by SOC loss by soil erosion.

Second, the restored forest showed a poor ability to react to external disturbance. This recovered soil C by ecological restoration would not be beneficial for long-term storage and may be more easily decomposed. For example, if both the primary forests and the restored forests are simultaneously destroyed, more soil C may be lost from the restored forests as they have lower C stabilization [24]. C releases resulting from deforestation have been considered one of the main sources of the increase

in atmospheric $CO_2$ concentration. However, current policies usually emphasize the protection of existing primary forests, i.e., preventing them from being deforested or destroyed [68,69]. Here, our results indicate that the vigorous development of vegetation restoration and the subsequent conservation of these restored forests are equally important for the global C balance and the mitigation of climate warming.

Third, in this study area, the forest stand structure mostly consists of *Pinus massoniana*. Xiong et al. [55] noted that forest composition made a minor contribution to soil C restoration. To increase SOC sequestration in soils and the long-term functional ability of forest ecosystems to act as C sinks, more broad-leaved trees, especially nitrogen-fixing trees, are needed [39]. Afforestation with mixed trees might be a better choice for SOC sequestration from a long-term perspective [24].

## 5. Conclusions

Assessing the restoration degree of degraded soils is critical for choosing appropriate restoration measures. Horizontal bamboo burl-groove can reduce soil erosion and SOC loss, while vegetation construction mainly plays a role as the source of SOC and nutrients. This study suggested that the combination measures of horizontal bamboo burl-groove and reforestation could accelerate the restoration of degraded soils, especially in relation to SOC recovery. The SOC stocks in restored lands under different recovery stages were much higher than those in eroded soils but still need a longer time to approach an equivalent level to nearby undisturbed forests. Long-term monitoring is necessary. The current study also showed that the active SOC fractions and SOC stocks, which are mutable and very sensitive and responsive to external influences, are good indicators for evaluating soil quality changes. The high efficiency of the combined measures suggested that horizontal bamboo burl-groove and continuous OC sources from vegetation are two main factors influencing SOC restoration. Combination measures of horizontal bamboo burl-groove and revegetation could be considered in restoring degraded slope lands in red soil areas. This study provided useful evidence-based information for governors to form management practices on eroded land resources and evaluate the C sink benefits of the comprehensive management of soil and water conservation.

**Author Contributions:** C.T. was responsible for funding acquisition and resources. S.X. conceptualized the research and resources. S.X., J.Z., J.D., H.L. performed the data curation and investigation. S.X. wrote the original draft. C.W. reviewed and edited the manuscript. All authors have read and agreed to the published version of the manuscript.

**Funding:** This work was supported by the National Key Research and Development Program of China (No. 2018YFC0407602), the National Natural Science Foundation of China (No. 41761063), and the Key Research Program of Jiangxi Provincial Water Conservancy Department (No. KT201716 and No. 201821ZDKT17).

**Acknowledgments:** The authors thank all anonymous reviewers for their helpful remarks. Special thanks are also expressed to Wang Yonglu for the field observations and Zhang Yongfen for help in drawing the study area map.

**Conflicts of Interest:** The authors have declared that no competing interests exist.

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
