# Peer review of "Soil Organic Carbon Sequestration and Active Carbon Component Changes Following Different Vegetation Restoration Ages on Severely Eroded Red Soils in Subtropical China"

_forests, doi:10.3390/f11121304_

Round 1

Reviewer 1 Report

The main question answered by the study is the restoration of soil organic carbon and active carbon components under the impact of soil erosion measures and reforestation following different restoration ages. The authors hypothesized that, with the constant influx of decomposing plant litter, long-term restoration of vegetation would increase carbon stores. The authors also assumed that the content of active organic carbon in soil gradually increases with increasing age of regeneration, which increases the risk of carbon decomposition in the soil and reduces the stability of its resources.

The research was conducted in the subtropical humid monsoon climate zone of the Pingjiang Basin of Jiangxi Province, which is located in the central mountain region of the red eroded soil region of southern China. The authors performed a study in bare land as a control and four restored forest sites with restoration ages of 4 years (F4), 14 years (F14), 24 years (F24) and 34 years (F34). A nearby-undisturbed forest (representing the climax vegetation) was also selected to test the recovered degree of the soil carbon stock after vegetation restoration.

Research on water erosion of soils and methods of its prevention is a very current and important topic. Currently, soil water erosion is considered to be one of the major soil degradation factors in the world. Therefore, anti-erosion protection is of particular importance as it prevents progressive degradation and allows to maintain the production potential of soils for past generations. A very important issue is also the reclamation and productification of soils degraded as a result of water erosion. In the article, the authors presented a method of restoring the content of organic carbon and activated carbon components in eroded red soil by applying mutually complementary technical and biological fortifications. The buildings used contributed to an increase in the carbon content in the soil. Increasing the content of organic carbon and active carbon components improves the soil’s fertility, which facilitates the growth of vegetation, which better protects the soil against erosion and leads to the productification of the degraded area. I believe that the research presented in this article is relevant and interesting.

The subject of the research is not original. The positive effect of vegetation on soil, including the increase in carbon-rich organic matter, is well known. However, such research is often conducted in completely different climate and soil conditions, with the use of different plant species and technical solutions. Therefore, it is often difficult to adapt ready-made solutions in other climatic zones. The research presented in the reviewed article is important from the possibilities point of view of using the technical and biological solutions presented in the article for soil reclamation in other areas with similar soil, climatic and topographic conditions, which have been degraded as a result of deforestation and the activation of erosion processes.

I believe that the article is well written. The point of the research is clearly stated. Descriptions of the research object, the assumed experience and the methodology used are complete and understandable. The test results are well documented and the discussion was conducted correctly. The text of the article is understandable and easy to read. The conclusions are consistent with the presented evidence and relate to the main goal of the research and the hypotheses. The authors proved that the combination measures of horizontal bamboo burl-groove and reforestation could accelerate the restoration of degraded soils, especially in relation to soil organic carbon recovery. The soil organic carbon stocks in restored lands  under different recovery stages were much higher than those in eroded soils but still need a longer  time to approach an equivalent level to nearby undisturbed forests.

All comments regarding the format of the article can be found in the attached file.

Author Response

Response to Reviewer 1 Comments

Dear reviewer:

Thank you for the relatively positive comments on this manuscript, however in recent years, studies regarding the influence of ecological restoration on soil C stocks have been conducted extensively. You highly recognized the effects of the combination measures of horizontal bamboo burl-groove and reforestation on soil SOC recovery, and the adaptability of the combination measures in other areas with similar soil, climatic and topographic conditions.The responses to your comments are showed below:

Point 1: the writing of references

Response 1: I am sorry for the not rigorous work attitude. I have made serious revision according to the “Forest-1007823-review”.

Point 2: in order to reduce the similarity index, the following expressions have been modified.

  • In line 70-72, “Afforestation is considered an effective method for alleviating the carbon dioxide (CO2) concentration and the pace of climate warming through the accumulation and long-term storage of C in both plant biomass and soil.”——“By the C accumulation and storage in both plant biomass and soil, vegetation restoration is an effective method to alleviate the carbon dioxide (CO2) concentration in the air.”.
  • In line 99-106, “From a chronosequence of adjacent forest sites, we addressed this issue in bare land (BL) as a control and four restored forest sites with restoration ages of 4 years (F4), 14 years (F14), 24 years (F24) and 34 years (F34). A nearby-undisturbed forest (UF, representing the climax vegetation) was also selected to test the recovered degree of the soil C stock after vegetation restoration. We hypothesized that, with the continued input of easily decomposed plant litter, long-term vegetation restoration would increase the soil C stocks. We also hypothesized the soil C stocks in the F34 sites would be lower than those in the UF sites because the restored forest age was still short relative to the age of the primary forests.” ——“From a chronosequence of adjacent forest sites, we addressed this research in bare land (BL) as a control and four restored forest sites with different restoration ages of 4 years (F4), 14 years (F14), 24 years (F24) and 34 years (F34). A nearby-undisturbed forest (UF, representing the climax vegetation) was also selected to test the recovered level of the soil C stock after long-tern vegetation restoration. With the continued input of plant litter, we hypothesized long-term vegetation restoration would increase the soil C stocks. We also hypothesized that, due to the restored forest age was still short relative to the age of the primary forests, the soil C stocks in the F34 sites would be lower than those in the UF sites.”.
  • In line 162-163, “In each plot, five core samples were randomly collected and combined into one composite sample for the different layers” ——“In each forest sites, five core samples were randomly collected and combined into one sample at different layers”.
  • In line 262-263, “The soil C stocks in 1 m depth soil significantly increased after afforestation when compared to the BL.”——“Compared to the BL, the soil C stocks in 1 m depth soil for restored forest sites significantly increased”.
  • In line 282-283, the sentence “Similar results have been documented in many previous studies” ——“Many previous studies have obtained similar conclusions”.
  • In line 284-285, “In this study, the increased soil C stocks after afforestation may be attributed to the input of plant residues into the soil following the establishment and growth of trees”——“In our study, the recovery of soil C stocks after vegetation restoration may be mainly due to the input of easy decomposed plant litter into the soil following the establishment and growth of trees, shrubs and herbs.”.
  • In line 297-298, “These inconsistent results may be primarily related to multiple factors, such as the degree of degradation, soil mineralogy, nutrient status”——“These inconsistent conclusions may be mainly due to different degradation degrees and soil nutrient levels”.
  • In line 356-358, “An earlier meta-analysis by Don et al. (2011) suggested that the increased C stocks in the afforested soils were still lower than those in the undisturbed forests”——“By meta-analysis, Don et al. (2011) suggested that soil C stock level after vegetation restoration were still lower than those in the undisturbed forests”.
  • In line 374, “may hence offset a small part of C accumulation in the soil”——“may hence offset a small part of recovered soil C”.
  • In line 383-392, “This sequestered soil C by afforestation would not be beneficial for long-term storage and may be more easily decomposed and returned to the atmosphere. For example, if both the primary forests and the afforested forests are simultaneously destroyed, more soil C may be lost from the afforested forests as they have lower C stabilization (Zhang et al., 2019). Carbon releases resulting from deforestation have been considered one of the main sources of the increase in atmospheric CO2 However, current policies usually emphasize the protection of existing primary forests, i.e., preventing them from being deforested or destroyed (Lal, 2005; Jacobs et al., 2015). Here, our results indicate that the vigorous development of afforestation and the subsequent conservation of these restored forests are equally critical for the global C balance and the mitigation of climate warming.”——“This recovered soil C by ecological restoration would not be beneficial for long-term storage and may be more easily decomposed. For example, if both the primary forests and the restored forests are simultaneously destroyed, more soil C may be lost from the restored forests as they have lower C stabilization (Zhang et al., 2019). C releases resulting from deforestation have been considered one of the main sources of the increase in atmospheric CO2 concentration. However, current policies usually emphasize the protection of existing primary forests, i.e., preventing them from being deforested or destroyed (Lal, 2005; Jacobs et al., 2015). Here, our results indicate that the vigorous development of vegetation restoration and the subsequent conservation of these restored forests are equally important for the global C balance and the mitigation of climate warming.”.

Reviewer 2 Report

The main purpose of the paper is clear and well resolved. The three hypotheses set were also properly refuted. Methods used in the study are correct and properly applied. The results could help to get light on the controversy about the progression of carbon sequestration rates on restored soils and ecosystems. And, personally, I consider the horizontal bamboo burl-groove strategy quite interesting.
The introduction is clear and offers to the readers a not alarming biased or reprehensible state of the art.
In my opinión, some ecological variables are missing in the study. Ecosystems functionality, robustness, and biochemical cycling rates are quite dependent on the biodiversity on it. Wild herbivorous are grazing on it? are the birds dispersing the seeds? are the insects helping to pollinate?. Are soil microbiomes healthy and biodiverse in each patch? I am sure that you only measure de biodiversity index of plants.
With more variables, you can also perform a more informative statistical analysis. For example, can a principal component analysis of the variables discriminate low carbon sequestration form high? What are the main variables affecting carbon fixation and recycling? Your study is just a case description, but you have data enough to try to explore the causality of the observations.
As a minor comment, I consider that pseudo-timeline studies benefit considerably from a more visual representation. XY plot with time BL, 4, 14, 24, 34, and UF in the abscissa and the observed variables in the ordinate ax will help the reader to build a quick idea of the relationship between variables and their progression.
Finally, you did not offer any data about soil microtopography. I consider that talk about the microtopography of the soil it in the conclusión is not appropriate without data that supports your conclusion.
For the rest, I consider it to be an honest, interesting, and useful job.
Thanks for it.

Author Response

Response to Reviewer 2 Comments

Dear reviewer:Thank you for your constructive comments of our manuscript. Your suggestion is very helpful for improving quality of this manuscript. We have revised the manuscript according to your comments which we hope meet with approval. The responses to your comments are showed below:

Point 1: In my opinion, some ecological variables are missing in the study. Ecosystems functionality, robustness, and biochemical cycling rates are quite dependent on the biodiversity on it. Wild herbivorous are grazing on it? Are the birds dispersing the seeds? Are the insects helping to pollinate? Are soil microbiomes healthy and biodiverse in each patch? I am sure that you only measure de biodiversity index of plants. 

With more variables, you can also perform a more informative statistical analysis. For example, can a principal component analysis of the variables discriminate low carbon sequestration form high? What are the main variables affecting carbon fixation and recycling?  Your study is just a case description, but you have data enough to try to explore the causality of the observations.

Response 1: We are very grateful for this suggestion. In this study, the SOC, aboveground plant diversity (Table 1) and some soil indicators (Table 2) were measured. However, the correlation analysis between these ecological variables and the change of SOC wasn’t detected.

In the revised version, we made the Pearson’s correlation analysis between soil OC variables (TOC, DOC, and MBC) and some soil properties. But what a pity, there is no supplementary data on soil microbial community, which is closely related to soil active OC, and needs to be strengthened in the later research.

Point 2: As a minor comment, I consider that pseudo-timeline studies benefit considerably from a more visual representation. XY plot with time BL, 4, 14, 24, 34, and UF in the abscissa and the observed variables in the ordinate ax will help the reader to build a quick idea of the relationship between variables and their progression. 

Response 2: In order to maintain consistency, I think it is better to set the sample plot under different recovery stages as the abscissa.

Point 3: Finally, you did not offer any data about soil microtopography. I consider that talk about the microtopography of the soil it in the conclusion is not appropriate without data that supports your conclusion.

Response 3: Thanks your suggestion. In order to improve the accuracy of the study, “suitable microtopography modification” has been changed to “horizontal bamboo burl-groove”.